# Hantavirus infection-induced B cell activation elevates free light chains levels in circulation

Jussi Hepojoki[1,2], Luz E. Cabrera[1☉], Satu Hepojoki[1☉], Carla Bellomo[3], Lauri Kareinen[1], Leif C. Andersson[4,5], Antti Vaheri[1], Satu Mäkelä[6,7], Jukka Mustonen[6,7], Olli Vapalahti[1,5,8], Valeria Martinez[3], Tomas Strandin[1]*

1 University of Helsinki, Medicum, Department of Virology, Helsinki, Finland, 2 University of Zürich, Vetsuisse Faculty, Institute of Veterinary Pathology, Zürich, Switzerland, 3 Laboratorio Nacional de Referencia para Hantavirus, Servicio Biología Molecular, Departamento Virología-INEI-ANLIS "Dr C. G. Malbrán," Buenos Aires, Argentina, 4 University of Helsinki, Medicum, Department of Pathology, Helsinki, Finland, 5 Helsinki University Hospital Laboratory, Helsinki, Finland, 6 Tampere University Hospital, Department of Internal Medicine, Tampere, Finland, 7 Faculty of Medicine and Health Technology, Tampere University, Tampere, Finland, 8 University of Helsinki, Veterinary Faculty, Veterinary Biosciences, Helsinki, Finland

☉ These authors contributed equally to this work.
* tomas.strandin@helsinki.fi

**Data Availability Statement:** All relevant data are within the manuscript and its Supporting Information files.

## Abstract

In humans, orthohantaviruses can cause hemorrhagic fever with renal syndrome (HFRS) or hantavirus pulmonary syndrome (HPS). An earlier study reported that acute Andes virus HPS caused a massive and transient elevation in the number of circulating plasmablasts with specificity towards both viral and host antigens suggestive of polyclonal B cell activation. Immunoglobulins (Igs), produced by different B cell populations, comprise heavy and light chains; however, a certain amount of free light chains (FLCs) is constantly present in serum. Upregulation of FLCs, especially clonal species, associates with renal pathogenesis by fibril or deposit formations affecting the glomeruli, induction of epithelial cell disorders, or cast formation in the tubular network. We report that acute orthohantavirus infection increases the level of Ig FLCs in serum of both HFRS and HPS patients, and that the increase correlates with the severity of acute kidney injury in HFRS. The fact that the kappa to lambda FLC ratio in the sera of HFRS and HPS patients remained within the normal range suggests polyclonal B cell activation rather than proliferation of a single B cell clone. HFRS patients demonstrated increased urinary excretion of FLCs, and we found plasma cell infiltration in archival patient kidney biopsies that we speculate to contribute to the observed FLC excreta. Analysis of hospitalized HFRS patients' peripheral blood mononuclear cells showed elevated plasmablast levels, a fraction of which stained positive for Puumala virus antigen. Furthermore, B cells isolated from healthy donors were susceptible to Puumala virus in vitro, and the virus infection induced increased production of Igs and FLCs. The findings propose that hantaviruses directly activate B cells, and that the ensuing intense production of polyclonal Igs and FLCs may contribute to acute hantavirus infection-associated pathological findings.

**Funding:** This study was supported by Academy of Finland (www.aka.fi) (grant numbers 308613 and 321809) to JH and TS, Competitive State Research Financing of the Expert Responsibility Area of Tampere University Hospital to JM, Sigrid Juselius Foundation to JM and AV, Magnus Ehnroot Foundation to AV, Foundation for Clinical Chemistry Research to SH, Laboratoriolääketieteen Edistämissäätiö to SH and Suomalais-Norjalainen Lääketieteen Säätiö to SH. The funders had no role in study design, data collection and analysis, decision to publish, or preparation of the manuscript.

**Competing interests:** The authors have declared that no competing interests exist.

## Author summary

Orthohantaviruses are globally spread zoonotic pathogens, which can cause hemorrhagic fever with renal syndrome (HFRS) and hantavirus pulmonary syndrome (HPS) with significant burden to human health. The pathogenesis mechanisms of orthohantavirus-caused diseases are not known in detail; however, excessive immune response towards the virus with concomitant pathological effects against host tissues appears to be a contributing factor. Here we report an increase of free immunoglobulin (Ig) light chains (FLCs), components required to make complete Ig molecules, in blood of acute HFRS and HPS. Samples collected during acute HFRS demonstrated increased FLCs levels in the urine and blood of patients hospitalized due the disease. Furthermore, the FLC levels positively correlated with markers of acute kidney injury. In addition, our results show that ortho-hantaviruses can infect and activate B cells to produce FLCs as well as whole Igs, which provides a mechanistic explanation of the increased FLC levels in patients. Taken together, our results suggest that aberrant antibody responses might play a role in the pathogenesis of orthohantavirus infections.

## Introduction

The pathogenicity of zoonotic viruses often links to the mechanisms the virus employs to fight host immune system [1]. Orthohantaviruses, a genus of rodent-, insectivore- and bat-borne viruses, cause a persistent and seemingly benign infection in their reservoir hosts [2]. Zoonotic transmission of rodent-borne orthohantaviruses occurs via aerosolized rodent excreta and may lead to hemorrhagic fever with renal syndrome (HFRS) or hantavirus pulmonary syndrome (HPS). The incubation period for HFRS ranges from ten days to six weeks and the infection starts with fever accompanied by headache, abdominal and back pains, nausea and vomiting. Increased vascular leakage can result in fluid accumulation in various tissues and sometimes hypotension and shock. Thrombocytopenia and leukocytosis are typical laboratory findings [3]. Acute kidney injury (AKI) is common in hospital-treated patients. Renal involvement includes oliguria, hematuria and proteinuria. Polyuric phase precede the recovery [4]. The severity of HFRS varies depending on the causative orthohantavirus, e.g., Hantaan virus (HTNV) causes a severe- and Puumala virus (PUUV) a mild HFRS with respective case fatality rates of ~5% and 0.08–0.4%. However, AKI is not a common finding in HPS, where fatality rates can exceed 40% and the most affected organ is the lungs [2].

The kidney pathophysiology of HFRS remains elusive. The marked proteinuria associated with HFRS is of both glomerular and tubular origin, indicating acute failure in glomerular sieving and tubular re-absorptive functions [4]. Histological analysis reveals that the glomerular apparatus remains relatively intact, but tubular epithelial cells can show even necrosis [4]. A common histopathological finding is acute tubulointerstitial nephritis (ATIN) with infiltration of various immune cells, and elevated levels of cytokines as well as adhesion molecules further imply an immune-mediated pathogenesis [5,6]. Orthohantaviruses target mainly endothelial cells lining the vasculature [3,7]. However, the virus also infects renal tubular epithelial cells, as demonstrated by histology of acute HFRS patients and experimentally infected macaques [8]. As the transmission to humans occurs via inhalation of aerosolized rodent excreta [2]. The lungs are the initial target organ of the virus and infection of endothelial cells lining the bloodstream is the probable route of spread to the kidneys and other parts of the body [2]. The reasons for the long incubation period between infection and appearance of symptoms remains unknown.

In a recent report, we described how PUUV infection gave rise to free light chains (FLCs) capable of binding viral N protein [9]. These findings prompted us to study whether overproduction of FLCs occurs in acute hantavirus infection, together with the possible association between excess FLCs and HFRS-associated AKI. Because circulating activated B cells presumably produce FLCs [10,11], we further analyzed the ability of PUUV to infect and activate B cells both *in vitro* and in patients.

## Materials and methods

### Ethics statement and clinical samples

The Ethics Committees of Tampere University Hospital (permit numbers 99256 and R04180) and Comité de Etica en Investigación—ANLIS "Dr. C. Malbrán" (FOCANLIS 2015–1532) approved the use of patient samples. All subjects gave a written informed consent in accordance with the Declaration of Helsinki. The study material consisted of peripheral blood mononuclear cells (PBMCs), serum, plasma, and urine from hospitalized, serologically confirmed acute PUUV infection at Tampere University Hospital, Finland, between September 2000 and March 2009. The samples were collected sequentially during hospitalization (acute stage and during recovery convalescent stage). Sequential samples included PBMCs from 13 as well as serum and urine from 14 patients (S1 Fig). Additionally, samples included serum collected on the first hospitalization day of 56 patients (S1 Table). As references, serum from patients with clinically suspected PUUV infection but serologically negative (from Helsinki and Uusimaa Hospital District laboratory HUSLAB) as well as healthy controls were included (n = 8 for both). In the case of HPS, serum samples from 54 cases (2–13 days after onset of fever) and 10 healthy controls were used (S2 Table). The HPS cases were categorized in terms of severity grades based on the classification employed in [12]: Grade 1, patients with prodromal symptoms without respiratory involvement; Grade 2, patients with mild to moderate respiratory compromise without hemodynamic compromise; Grade 3, patients with severe respiratory insufficiency with hemodynamic compromise; Grade 4, patients with severe respiratory insufficiency with refractory-to-treatment hemodynamic compromise and fatal outcome. PBMC were stored in -135˚C, blood and urine samples in -80˚C prior to analysis.

The study included archival Bouin-fixed, paraffin-embedded kidney biopsies collected during 1978–1989 at Tampere University Hospital (ethical permit number R18808, S3 Table). The highest measured serum creatinine level of the patients ranged from 220 to 1,050 μmol/L. Creatinine was determined by Cobas Integra (F. Hoffmann-La Roche Ltd.). National supervisory authority of Health and Welfare approved the study on kidney biopsies (No. V/19454/2018).

### Protein quantification

Immunoglobulin Free Light Chains Kappa and Lambda Human ELISA (Biovendor) served for measuring FLCs in serum, plasma and urine of all study patients (all 1:200 dilution) and in B cell culture supernatants (1:2 dilution for lambda, 1:5 for kappa). ELISA kits obtained from Mabtech served for measuring total IgA, IgM and IgG levels from PUUV patient plasma and from B cell culture supernatants. BCA assay kit (Thermo Scientific) served for measuring total urinary protein solubilized in 5% SDS in 0.5 N NaOH after 10% trichloroacetic acid precipitation and 70% ethanol washes. The determination of urine albumin (U-Alb) was made by an immunoturbidometric method on a Cobas C 702 –clinical chemistry analyzer (F. Hoffman–La Roche Ltd.). The excretion rate in the urine was calculated using the volume of urine collected in a recorded timeframe (minutes).

## Immunohistochemistry

Paraffin-embedded kidney biopsies of patients with acute HFRS were stained with rabbit poly-clonal κ or λ LC-specific antibodies (Dako) followed by polymer-based horseradish peroxidase (HRP) chromogenic detection and hematoxylin counterstain. Biopsies from patients with unrelated kidney disease served as the controls. The staining followed standard procedures of Helsinki University diagnostic laboratory HUSLAB. Image digitalization was done with Zeiss Axioscan slide scanner and enumeration of κ or λ LC-positive cells in relation to total cell number was performed using Aiforia v4.8 cloud platform (Aiforia Technologies Oy, Helsinki, Finland). The platform employs image-based artificial intelligence (AI) with high-performance cloud computing to discriminate between HRP-positive and -negative cells in individual tissues. The AI training process included ~2500 iterations based on ~500 annotations of manually selected positive and negative objects with visible nuclei.

## Blood fractionation and B cell enrichment

Ficoll-Paque (GE Healthcare) served in isolation of peripheral blood mononuclear cells (PBMCs) from EDTA-anticoagulated blood of PUUV patients and healthy volunteers or from buffy coats obtained from the Finnish Red Cross Blood Service. The PBMC isolations followed the protocol of Miltenyi Biotec. The obtained PBMCs were either frozen in -135˚C (as was the case for all PUUV patient samples), used directly for *in vitro* infection experiments with PUUV, or subjected to B cell enrichment through negative selection by EasySep™ Human Pan-B Cell Enrichment Kit (Stemcell technologies). The enrichment routinely resulted in ~90% purity of $CD19^+$ B cells, as determined by flow cytometry.

## Cell lines and viruses

PUUV (strain Kazan) was propagated in Vero E6 cells (African green monkey kidney epithelial cell line; ATCC no. CRL-1586) grown in Minimum Essential Medium Eagle (MEM; Sigma Aldrich) supplemented with 10% inactivated fetal calf serum (FCS), 100 IU/ml of penicillin, 100 μg/ml of streptomycin and 2 mM of L-glutamine. For infection experiments, the virus was pelleted by ultracentrifugation (SW28 rotor, 27,000 rpm, 50 min, +4˚C) through a 0.22 μm-filtered 30% ultra-pure sucrose cushion (in PBS), to obtain concentrated virus preparation free of cell culture contaminants. The pellets suspended into Vero E6 cell growth medium showed infectious PUUV titers of approximately $10^7$ fluorescent focus-forming units (FFU)/ml as described [6]. Where indicated, PUUV was inactivated with UV (300,000 μJ/$cm^2$, Stratalinker, Stratagene). For mock infections, an equal amount of conditioned cell culture supernatant was subjected to ultracentrifugation and the pelleted material treated as described for PUUV pellets.

## *In vitro* infection experiments

PBMCs or enriched B cells were inoculated with mock medium, with purified PUUV, or UV-inactivated PUUV (PUUV-UV) at multiplicity of infection (MOI) of 5, and incubated 5 days at 37˚C, 5% $CO_2$ in B cell growth medium. The B cells were maintained in RPMI-1640 (Sigma Aldrich) supplemented with 10% inactivated FCS, 100 IU/ml of Penicillin and 100 μg/ml of Streptomycin, 2 mM of L-glutamine, 20 ng/ml rIL-2 (recombinant interleukin-2, Immuno-tools) and ITS cell culture supplement (Sigma-Aldrich). The cells were pelleted by centrifugation (400 x g, 10 min) and PBMCs subjected to flow cytometry analysis (described below) and enriched B cells for immunofluorescence and western blot analyses.

In the case of ELISA and Elispot experiments, B cells, enriched from PBMCs by negative selection, were inoculated (1 h at 37˚C, 5% $CO_2$) with mock medium, PUUV, PUUV-UV, Sendai virus, or Sindbis virus at MOI of 1, all diluted in B cell growth medium. Non-infected cells with added CpG (1 μM) served as positive control for B cell activation. The virus inoculum was removed by pelleting the cells by centrifugation (400 x g, 10 min), and one million cells/ml were incubated in B cell growth medium at 37˚C 5% $CO_2$, for the times indicated in the figure legends. After incubation, the cells were pelleted by centrifugation (400 x g, 10 min), supernatants collected for the determination of Igs by ELISA, and the cells processed for Elispot analysis.

## Immunofluorescence

B cells were attached to glass slides using cytospin (Shandon), fixed with 4% paraformaldehyde (in PBS) for 10 min at RT, and stained for immunofluorescence detection of PUUV N protein, with the use of a rabbit polyclonal anti-PUUV N-specific serum, followed by AlexaFluor594-conjugated secondary Ab (ThermoScientific). Light chains were stained with polyclonal goat anti-κLC and anti-λLC antibodies (2 μg/ml each; Southern Biotech), followed by AlexaFluor488-conjugated secondary Ab (ThermoScientific). Cell nuclei were stained with Hoechst 33342.

## Western blot

B cells were lysed in non-reducing Laemmli sample buffer, subjected to sodium dodecyl sulfate-polyacrylamide gel electrophoresis and transferred to nitrocellulose membrane by standard procedures. The blots were sequentially probed with mouse monoclonal Ab (5E1) detecting PUUV N protein, rabbit anti-actin Ab and goat anti-κLC followed by appropriate IRdye680- or IRdye800-conjugated secondary Abs (Li-Cor). The blots were visualized using Odyssey infrared imaging system (Li-Cor).

## Flow cytometry

Frozen PUUV patient PBMCs were thawed in a 37˚C water bath for 5 min, added dropwise to RPMI-1640 (Sigma Aldrich) supplemented with 10% inactivated FCS, 100 IU/ml of penicillin, 100 μg/ml of streptomycin, 2mM of L-glutamine, and 100 μg/ml DNAse I (Sigma Aldrich). After a 10 min incubation at RT, the cells were washed once with the conditioned medium and once with PBS-EDTA (PBS with 1 mM EDTA), followed by cell quantification using Bio-Rad cell counter TC20.

One to three million PBMCs or 250,000 B cells (enriched from PBMC fraction, see above) were stained with fixable viability stain (FVS570, BD) for 15 min at RT. Then 1% FCS and FcR Blocking reagent (Immunostep) were added, and the cells stained for 30 min at RT with a cocktail of fluorescent-dye conjugated anti-human mouse mAbs recognizing cell surface antigens. The cocktail included antibodies against: CD3 FITC (clone OKT3; Thermo Scientific), CD56 FITC (B-A19) and CD66 FITC (6g5j; both from Immunotools), CD138 APC-Vio770 (44F9; Miltenyi), CD14 FITC (M5E2), CD38 PE-CF594 (HIT2), CD27 V500 (M-T271), HLA-DR BV786 (G46-6) and CD19 BV711 (SJ25C1; all from BD). For some samples, CD19 PE-Cy5 (HIB19; Thermo Scientific) and IgD BV605 (IA6-2; BD) were used. The cells were fixed, permeabilized (eBioscience fixation and permeabilization buffer; Thermo Scientific), and stained with mAbs against intracellular antigens for 30 min at RT. The following mAbs were used: anti-kappa light chain PercP-Cy5.5 (G20-193), anti-lambda light chain BV605 (JDC-12), anti-IgM APC (G20-127), anti-IgG AF700 (G18-145; all from BD) and anti-IgA PE-Vio770 (IS11-8E10; Miltenyi). In the experiments with AF647-conjugated bank vole mAb

recognizing PUUV N protein (1C12; [13,14]) the anti-IgM APC was replaced by anti-IgM V450 (G20-127; BD). After staining, the cells were washed with PBS-EDTA and fixed with 1% paraformaldehyde before FACS analysis with a three-laser (405, 488 and 640 nm) LSRII Fortessa instrument (BD).

The FACS data was analyzed with FlowJo 10.6. The datasets were normalized independently, concatenated and Uniform Manifold Approximation and Projection for Dimension Reduction (UMAP) was performed for the unsupervised identification of cell populations and dimensionality reduction [15], based on the detection of different fluorochromes used in the flow cytometry panel. To generate UMAP plots, the minimum distance was set at 0.5 and the nearest neighbors' distance was set at 15, using a Euclidean vector space. Clusters were then identified and described by performing a flow self-organizing map (flowSOM) [16] with a number of meta-clusters set to 10.

## Elispot

Elispot-assays, performed as described [17], served to detect IgA, IgM and IgG secreting cells. Briefly, nitrocellulose filter plates (Millipore) were coated with polyclonal goat anti-κLC and anti-λLC antibodies (10 μg/ml in PBS each; Southern Biotech) overnight at 4˚C. The plates were blocked (2% skimmed milk in RPMI-1640) at RT for 1 h, before adding *in vitro* infected washed B cells, at a 1:5 dilution in RPMI-1640 containing 0.5% BSA. After incubation (3 h at 37˚C), the plates were washed three times with PBS and four times with PBS-T (PBS+0.05% Tween 20). The plates were then incubated (1 h at RT) with polyclonal (all goat, Southern Biotech) AP-conjugated anti-IgA, HRP-conjugated anti-IgM, or HRP-conjugated anti-IgG antibodies, washed three times with PBST and three washes with MilliQ water. IgA secreting cells were detected using 5-Bromo-4-chloro-3-indolyl phosphate ρ-toluidine and nitroblue tetrazolium chloride (BCP/NBIT) "ready to use" solution (Sigma Aldrich) and IgM and IgG secreting cells using 3-amino-9-ethyl-carbazole (AEC). The AEC solution consisted of 400 μg/ml AEC (Sigma Aldrich; 10 mg/ml stock solution in dimethylformamide) and 0.012% $H_2O_2$ in 0.1 M acetate buffer (pH 5.0). AiD Elispot reader (AutoImmun Diagnostika) served for plate imaging and spot counting.

## Statistical analyses

GraphPad Prism 8.3 served to identify statistically significant differences between the treatment groups. The specific tests used are indicated in the figure legends. General estimating equations on SPSS v25 (IBM) was used to assess differences between sequential samples (the acute vs. convalescent phase).

# Results

## Circulating FLCs increase in acute hantaviruses diseases

We recently reported that patients with acute PUUV infection possess FLCs specific to PUUV N protein [9]. To study if hantavirus infection induces an overall increase in circulating κ and λ FLC concentrations, we used ELISA to compare the serum samples of patients hospitalized due to acute HFRS caused by PUUV and HPS caused by ANDV to those of patients suffering from other febrile illness (OFI) and healthy volunteers. Both HFRS and HPS patients showed significantly higher median concentrations of $\kappa_{FLC}$ and $\lambda_{FLC}$ ($\kappa_{FLC}$ = 34.05 and 10.66; $\lambda_{FLC}$ = 26.37 and 22.39 μg/ml, respectively) as compared to healthy volunteers ($\kappa_{FLC}$ = 4.81 and $\lambda_{FLC}$ = 9.96 μg/ml) or patients with PUUV-suspected but serologically negative febrile illnesses ($\kappa_{FLC}$ = 7.35 and $\lambda_{FLC}$ = 17.14 μg/ml). While the $\lambda_{FLC}$ levels remained comparable, the HFRS

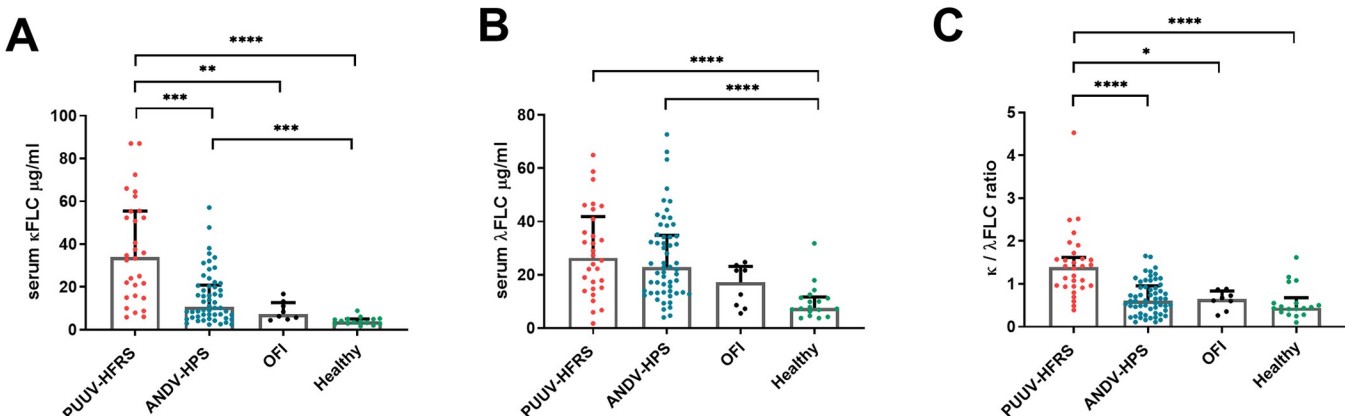

**Fig 1. Soluble FLCs increase in hantavirus diseases.** The concentrations of FLCs ($\kappa$ in **A** and $\lambda$ in **B**) as well as their ratio (in **C**) were measured by ELISA from serum samples of acute PUUV-caused HFRS (n = 30), ANDV-caused HPS (n = 50), other febrile illness (OFI; n = 8) and healthy volunteers (n = 18). The medians of patients with acute PUUV infection and OFI were compared to medians of healthy volunteers by Kruskal-Wallis + Dunn's multiple comparisons test and significant differences reported as * = p < 0.05, ** = p < 0.01, *** = p < 0.001 and **** = p < 0.0001. The bars indicate medians + interquartile ranges.

patient sera showed a significant increase in the $\kappa_{FLC}$ levels as compared to HPS patient sera (Fig 1A and 1B). The higher $\kappa_{FLC}$ levels further associated with significantly elevated $\kappa_{FLC}/\lambda_{FLC}$ ratios (median 1.39) in HFRS cases as compared to other sample groups (median ratios 0.44–0.64; Fig 1C).

## The levels of circulating FLCs associate with acute kidney injury in HFRS

Given that the serum $\kappa_{FLC}/\lambda_{FLC}$ ratio increases due to impaired renal FLC clearance [18,19], we studied if FLCs play a role in AKI by comparing the $\kappa_{FLC}$ and $\lambda_{FLC}$ levels to the maximum serum creatinine levels measured during the hospital stay in patients with acute HFRS (1st day of hospitalization). Indeed, both $\kappa_{FLC}$ and $\lambda_{FLC}$ levels positively correlated with the maximum creatinine values (Fig 2A and 2B), suggesting that FLC levels could serve as prognostic indicator in estimating the severity of AKI in PUUV-HFRS. The FLC levels correlated significantly also with urinary albumin excretion but not with the extent of thrombocytopenia or with the level of inflammation as indicated by maximum level of blood leukocyte count or maximum interleukin (IL)-6. However, the $\kappa_{FLC}/\lambda_{FLC}$ ratio did not significantly associate with serum creatinine or any other of the analyzed variables (S2A Fig). In addition, HFRS patients with the need of dialysis demonstrated significantly higher serum $\kappa_{FLC}$ levels in than patients not requiring dialysis (S3A Fig). In HPS, however, the amounts of serum $\kappa_{FLC}$ and $\lambda_{FLC}$ did not associate with disease severity, as assessed by the level pulmonary and hemodynamic involvement (S3B Fig).

To study the kinetics of FLC upregulation in HFRS, we measured their concentrations in serum samples collected sequentially during hospitalization, and at convalescence. The results displayed elevated median values of both FLCs in the acute stage, i.e. four to ten days after onset of fever (aof), as compared to samples drawn at recovery (grouped as 11–30 and 31–50 days aof) (Fig 2C). Similarly to FLC levels, the serum creatinine levels remained high throughout the acute stage up to 10 dpi (Spearman R = 0.824, p < 0.0001, n = 45 for $\lambda_{FLC}$; Spearman R = 0.836, p < 0.0001, n = 45 for $\kappa_{FLC}$; Fig 2D). At the same time, $\kappa_{FLC}/\lambda_{FLC}$ ratio showed high levels in the early acute stage that gradually declined towards recovery (Fig 2D). The highest deviations in the levels of circulating leukocytes and thrombocytes (the highest and the lowest, respectively), as compared to normal range, were also observed in the early acute phase

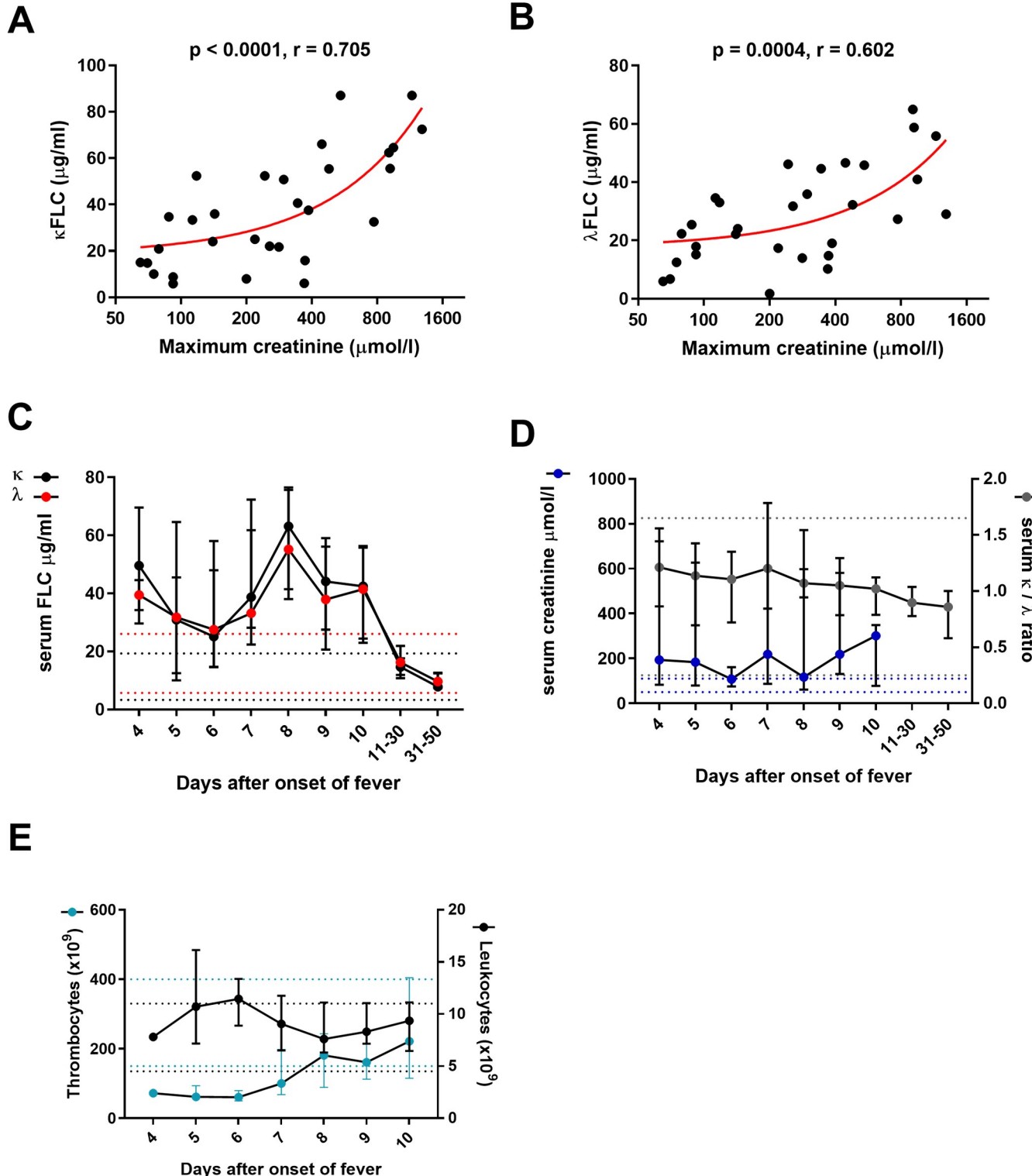

**Fig 2. Soluble FLCs correlate with the extent of kidney injury in HFRS.** (**A** and **B**) To assess the prognostic value of serum FLC levels for acute kidney injury, the concentration of FLCs (κ in **A** and λ in **B**) in serum samples collected at first day of hospitalization from patients with acute PUUV-HFRS (n = 30) were correlated to the maximum serum creatinine values measured during the hospital stay, using Spearman's rank correlation coefficient. Non-linear association between parameters is depicted by the red line. (**C**) The serum concentration of FLCs were measured in sequential serum samples obtained from patients hospitalized due to PUUV-HFRS (total patient n = 13, n for individual time points depicted in S1 Fig) in acute stage (days 4–10 after onset of fever) and convalescence (grouped as 11–30 and 31–50 days after onset of fever). From the same samples as in (**C**), two indicators of renal clearance function, serum

creatinine and serum κ and λ (κ / λ) FLCs ratio, were plotted together in (**D**) as well as thrombocyte and leukocytes counts in (**E**). The medians ± interquartile ranges are shown in (**C, D, E**). The dotted lines indicate normal range limits for different parameters (24).

(Fig 2E). The data further suggest a strong link between the severity of AKI and the circulating levels of both $\kappa_{FLC}$ and $\lambda_{FLC}$. Instead, the $\kappa_{FLC}/\lambda_{FLC}$ ratio and the severity of AKI showed no correlation, suggesting that factors other than impaired renal clearance contribute to the increased circulating FLCs levels during acute HFRS.

## Kinetics of urinary FLCs during the course of PUUV-HFRS

To evaluate the renal clearance of FLCs during HFRS, we measured the urinary FLC levels in samples collected from patients with acute PUUV infection during hospitalization and in recovery. The highest urinary output of both $\kappa_{FLC}$ and $\lambda_{FLC}$ occurred in early acute stage and was followed by gradual decline to normal, barely detectable, levels at recovery (>10 days aof, Fig 3A). Interestingly, the $\lambda_{FLC}$ but not the $\kappa_{FLC}$ levels significantly followed the total urinary protein output in urine (Spearman R = 0.33, p = 0.019, n = 51 for $\lambda_{FLC}$; Spearman R = 0.19, p = 0.343, n = 51 for $\kappa_{FLC}$, S4 Fig), suggesting that proteinuria alone does not explain the urinary $\kappa_{FLC}$ level increase. Comparison of the serum and urine FLC levels over the course of HFRS revealed a peak in urine FLC levels in the early acute stage (4–6 days aof, Fig 3B and 3C) whereas a peak in serum FLC levels occurred in late acute stage (7–10 days aof, Fig 3B and 3C). The differential kinetics of urine vs. serum FLC levels was further highlighted by the absence of correlation between urine and serum $\kappa_{FLC}$ or $\lambda_{FLC}$ levels (Spearman R = -0.132, p = 0.430, n = 38 for $\kappa_{FLC}$ and R = -0.049, p = 0.796, n = 38 for $\lambda_{FLC}$).

## Elevated renal plasma cell infiltration during acute PUUV-HFRS

Abnormally elevated serum FLC levels and renal insufficiency often associate with plasma cell disorders [20]. The identification of increased FLC levels in serum and urine of PUUV-HFRS prompted us to study the possibility of plasma cell infiltration in patient kidneys. Therefore, we performed immunohistochemistry using anti-κ and anti-λ chain antibodies on kidney biopsies of acute PUUV-HFRS and unrelated renal disease patients to demonstrate the presence of plasma cells (S3 Table). The staining revealed significantly increased numbers of infiltrating plasma cells of both κ (3.21% vs. 2.29% of all cells) and λ (1.71% vs. 0.73% of all cells) LCs in acute PUUV-HFRS as compared to control patients (Fig 4A and 4B). Furthermore, the

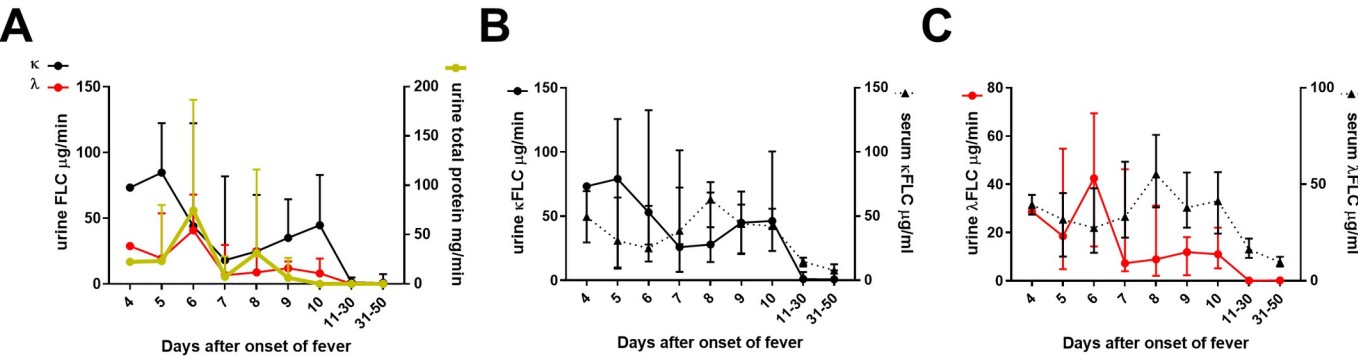

**Fig 3. FLCs increase in urine in patients with acute PUUV-HFRS.** (**A**) The concentration of FLCs and total protein levels were measured in sequential urine samples from the same patients as in Fig 2A and 2B. The protein concentrations in urine was normalized based on total urinary volume produced during an overnight ~8h collection. The kinetics of κFLC (in **B**) and λFLC (in **C**) secretion in urine and in circulation (serum samples, same data as in Fig 2A) were compared. The medians ± interquartile ranges are shown.

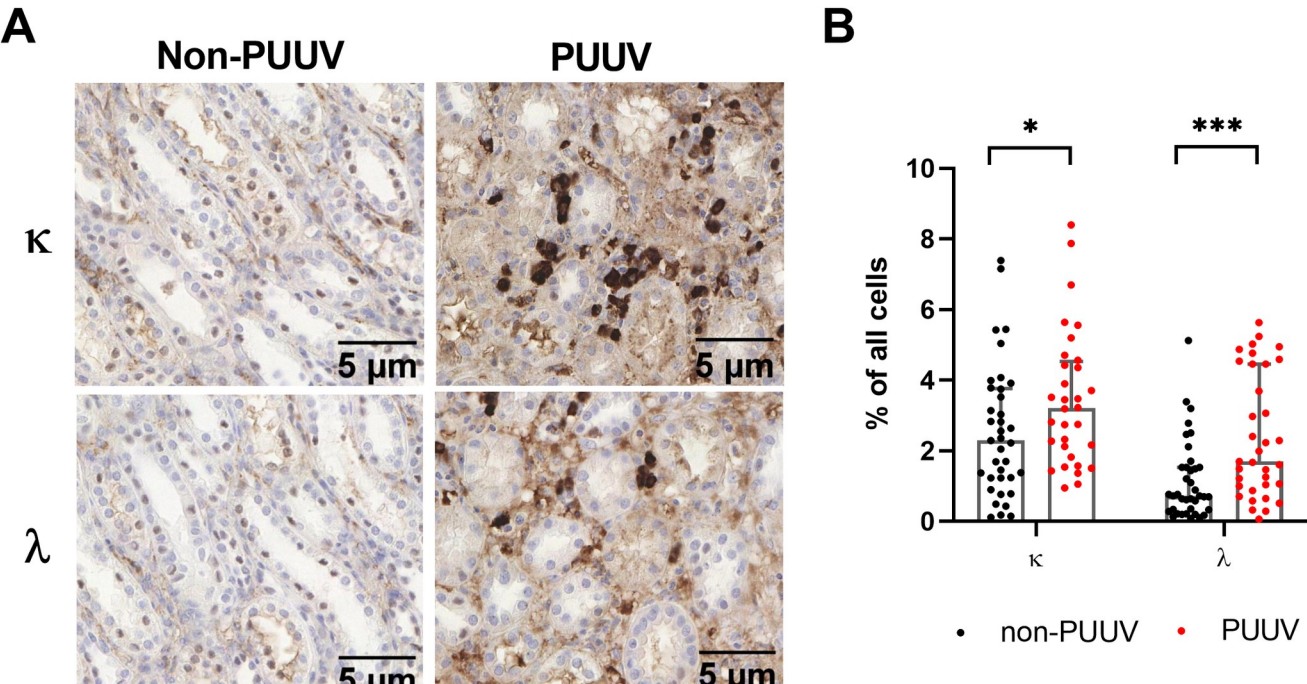

**Fig 4. Plasma cells infiltrate kidneys during acute PUUV-HFRS.** Sections from archival kidney biopsies of acute PUUV-HFRS or from patients with a kidney disease of another etiology were stained for κ or λ light chains by immunohistochemistry. (**A**) Representative images of κ and λ light chain staining of a patient with acute PUUV-HFRS and one patient with another cause of kidney dysfunction (glomerulonephritis) as control, showing the presence of infiltrating plasma cells in kidneys during acute PUUV-HFRS. (**B**) Quantification of the number of plasma cells in non-PUUV and acute PUUV-HFRS (n = 37–38 for non-PUUV controls and 32–35 for PUUV-HFRS). The medians across groups were compared by Mann-Whitney test and significant differences reported as * = p < 0.05 and *** = p < 0.001. The bars indicate medians + interquartile ranges.

infiltrated plasma cells in PUUV-HFRS predominantly localized to the interstitial space (Fig 4A), which is consistent with the typical diagnosis of acute tubulointerstitial nephritis in PUUV-HFRS (S3 Table). These findings point out to strong localized plasma cell response in acute PUUV infection, which could contribute to kidney clogging by inducing high local Ig and FLC concentrations. In addition, local production of FLCs in the kidneys could explain the lack of correlation between urinary FLC and serum FLC levels. The number of plasma cells in the kidneys positively correlated with the number of CD68+ and CD14+ cells (S2B Fig), which were previously analyzed from the same sample set [21] and likely represented macrophages and infiltrating monocytes, respectively. These results suggest a direct link between plasma cells and general inflammatory response in the kidneys during acute HFRS.

## Elevated levels of circulating plasmablasts (PBs) in acute PUUV

The observed infiltration of plasma cells into the kidneys suggests an extensive B cell activation during acute PUUV-HFRS. An increase in the amount of circulating PBs, the plasma cell precursors, occurs in ANDV-caused HPS [12]. Hence, we employed multiparametric flow cytometry to assess the abundance of PBs in PBMCs collected over the course of PUUV-HFRS from hospitalization to convalescence. Cells representing other than B cell lineage (CD3, CD14, CD56 and CD66) were removed through gating (S5 Fig) prior to dimensionality reduction with the use of unsupervised clustering methods (UMAP and FlowSOM). Cells clustered into 10 populations, of which seven represented CD19+ B cells. The frequencies of the various B cell populations differed dramatically in acute (4–10 aof) vs. convalescent (11–50 aof) phase of

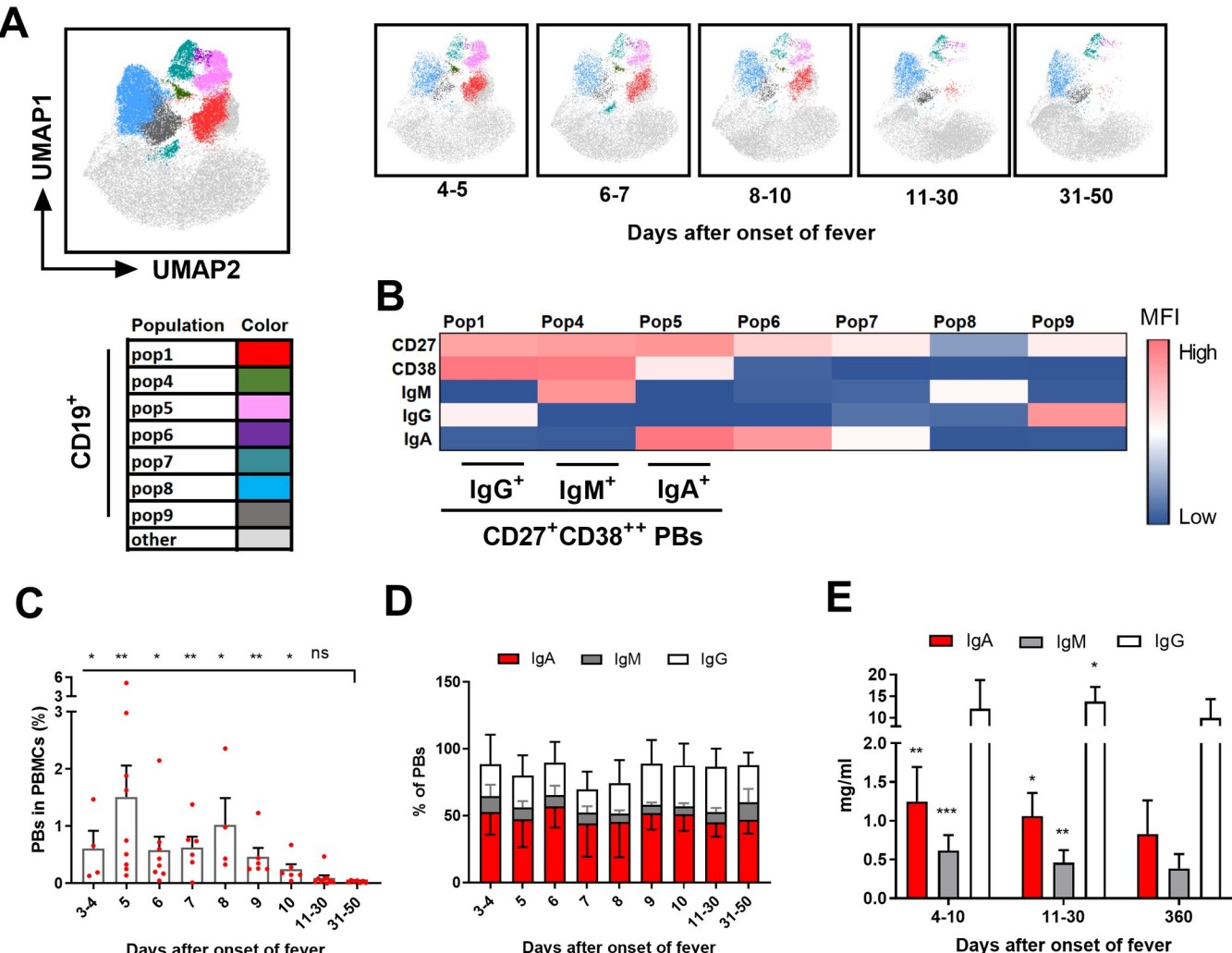

**Fig 5. Plasmablast frequencies increase in the circulation of patients with acute PUUV-HFRS. A)** UMAP and FlowSOM analysis was used to identify different cell populations in single live PBMCs, from which CD3, CD14, CD56 and CD66 positive cells were removed by conventional gating (gating strategy shown in S2 Fig). The distinct CD19[+] cell populations are shown by specific colors, whereas the CD19[-] populations are merged as light grey color in the plot. (**B**) Heatmap showing the relative mean MFI of CD27, CD38, IgM, IgG and IgA in different CD19[+] cell populations. (**C**) The frequencies of CD19[+] PBs out of total number PBMC used for flow cytometry analysis (gating shown in S2 Fig) were calculated for each sample at indicated days post onset of fever. Data is obtained from 13 PUUV-HFRS patients. (**D**) The frequencies IgA[+], IgM[+] and IgG[+] PBs in relation to the total number of PBs for each sample were calculated and plotted similarly as in **C**. (**E**) EDTA-plasma from acute (1st day of hospitalization, 5–8 days post onset of symptoms), early recovery (15–30 days post onset of symptoms) and full recovery (180 days post onset of symptoms) were subjected to total IgA, IgM and IgG ELISA (n = 14 for all groups). Significant differences between each indicated time point to full recovery (31–50 days post onset of symptoms in **C** and 180 days post onset of symptoms in **E**) were calculated using generalized estimating equations. *$p < 0.05$, **$p < 0.01$, ***$p < 0.001$.

PUUV-HFRS (Fig 5A). Based on cell surface expression of CD27 and CD38 markers, as well as intracellular Ig isotype expression, three populations were identified as PBs (pop1 = IgG[+] PBs, pop4 = IgM[+] PBs and pop5 = IgA[+] PBs), whereas the other four populations probably represented naïve and memory B cell subsets (Fig 5B). Next the abundance of CD19[+]/CD27[+]/CD38[+++] PBs (S5 Fig shows the gating strategy for representative acute and recovery phase samples) was normalized to the total number of PBMCs in the sample (Fig 5C). We observed a significant expansion of PBs in the acute stage (4–10 days aof) as compared to the samples drawn at recovery (at 31–50 days aof). The median frequency of PBs fluctuated extensively during the acute stage from 0.5% to 1.5% of all PBMCs. These findings indicate a marked B

cell activation during acute PUUV-HFRS, paralleling the observations on acute HPS patients [12].

Further analysis revealed the acute-phase PBs to be distributed as ~55% being IgA[+], ~35% IgG[+], and ~10% IgM[+], with no statistically significant differences in their relative frequencies between acute-phase and recovery samples (Fig 5D). However, lower levels of IgM[+] PBs (~5%) were present at the late acute phase (10 and 11–15 days aof) versus convalescence (~15%, 31–50 days aof). The distribution of plasmablast Ig isotypes during PUUV infection was similar to healthy controls [22], indicating a polyclonal "innate"- type B cell activation rather than expansion of virus-specific B cell clones. The findings further demonstrate that IgA[+] and IgM[+] PBs are relatively more abundant than IgG[+] PBs in the early acute phase as compared to early and full recovery. To estimate the impact of PB expansion (i.e. B cell activation) on Ig production, we measured the concentrations of circulating IgA, IgM, and IgG. The acute-phase samples presented significantly elevated IgA and IgM levels, whereas the samples taken at early recovery demonstrated elevated IgG levels, as compared to full recovery samples (Fig 5E).

## PUUV infects B cells

Hantaviruses infect microvascular endothelial cells and macrophages in patients [3,23], but their ability to infect B cells is currently unclear. To investigate if PUUV can directly infect B cells, we inoculated PBMCs isolated from healthy volunteers with either "live" (PUUV) or UV-inactivated PUUV (UV-PUUV). Flow cytometry analysis revealed PUUV N protein (PUUV-NP) expression in both naïve (IgD+ CD27-) and memory (IgD- CD27+) B cells at 2 and 5 days post infection (dpi) (the gating strategy shown in Fig 6A). The PUUV- and UV-PUUV-inoculated PBMCs analyzed at 2 dpi did not show statistically significant differences in the amount of PUUV-NP[+] cells (Fig 6B). However, analysis at 5 dpi showed significantly more PUUV-NP[+] naïve B cells in PUUV, in contrast to UV-PUUV infected PBMCs (Fig 6B), indicating virus replication. In addition, the number of PUUV-NP[+] memory B cells appeared to be elevated, although the difference was not statistically significant. To provide additional evidence of the B cells' ability to support PUUV replication, we inoculated enriched B cells of healthy volunteers (enrichment by negative selection resulted in ~90% B cell purity as assessed by CD19 expression, S6A Fig) with PUUV and UV-PUUV. Fluorescence microscopy revealed PUUV-NP expression in LC-positive B cells inoculated with PUUV, but not in those inoculated with UV-PUUV (Fig 6C), confirming successful that PUUV can replicate in B cells. Western blot served to assess productive PUUV replication, and showed increased PUUV NP expression in enriched B cells inoculated with live PUUV versus UV-PUUV (S6B Fig). Interestingly, western blot analysis also revealed increased production of FLCs by PUUV, as compared to UV-PUUV, in B cells.

To study whether PUUV infection of B cells occurs *in vivo*, we analyzed PBMCs collected at the acute and recovery phase of PUUV-HFRS using flow cytometry. Following the gating strategy described in Fig 5A, with the exception of intracellular PUUV-NP staining instead of whole Igs (gating for representative samples shown Fig 6D), we determined the mean frequencies of PUUV-NP[+] CD19[+] (B cells) and CD19[+]CD38[+++] cells (PBs). PUUV-NP[+] cells during the early acute stage displayed the highest cell counts (~0.6% out of CD19[+] and ~0.5% of CD38[+++] cells at 2–5 days aof), which declined towards recovery and reached undetectable levels at 30 days aof (Fig 6E). Interestingly, the data showed PUUV-NP[+] cells to be almost exclusively PBs (CD38[+++] cells, Fig 6D and 6E). As shown earlier (Fig 5), approximately 30% of CD19[+] cells in the acute-phase samples (from 4–9 days aof) were CD38[+++] (PBs), and the amount of PBs returned to baseline levels (< 10%) at 30 days aof (Fig 6E). Similarly to the in vitro infection experiments, PUUV-NP expression in patient B cells was detected also by

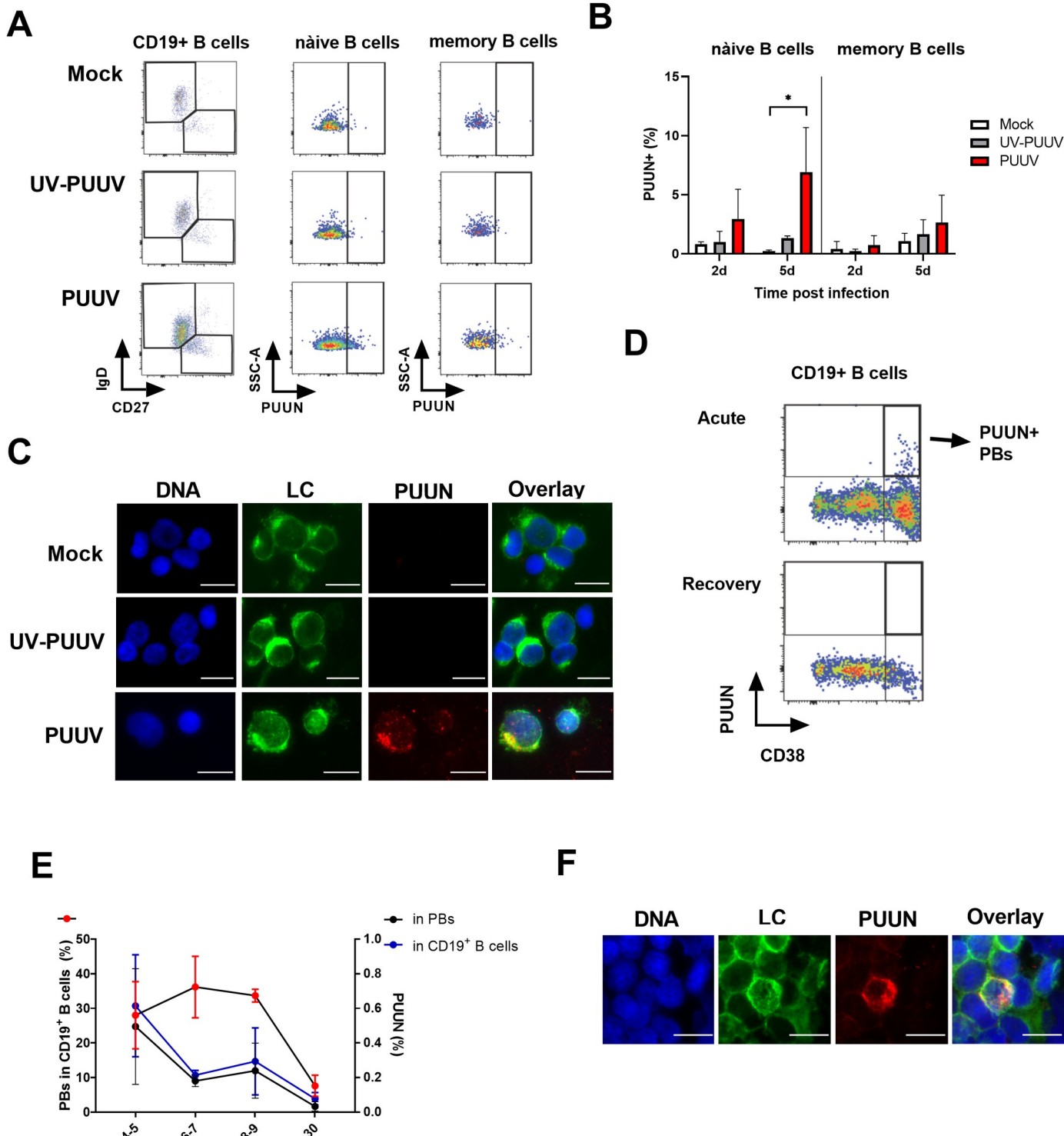

**Fig 6. PUUV infects B cells.** (**A**, **B**) PBMCs from healthy volunteers were inoculated with live PUUV or UV-inactivated PUUV as a control at a multiplicity of 5 FFFU virus/PBMC and cultured for 2 or 5 days. Naïve (CD19+IgD+CD27-) and memory (CD19+IgD-CD27+) B cells were identified by multi-parametric flow cytometry and, as an indication of PUUV infection, intracellular staining for PUUN using PUUN-specific AlexaFluor647-conjugated Mab 1C12 was included. (**A**) An example of the gating strategy to detect PUUN+ naïve or memory B cells is shown. (**B**) The mean frequencies of PUUN+ cells out of total number of naïve or memory B cells in inoculated PBMCs were compared between non-treated and PUUV or UV- PUUV inoculated groups with two-way ANOVA and Dunnett's multiple comparison tests. *p < 0.05, **p < 0.01, ***p < 0.001. (**C**) B cells were isolated from healthy volunteer PBMCs, inoculated with live PUUV or UV-PUUV

and cultured for 5 days at 37˚C. Cells were spun down on glass slides and fixed, permeabilized and stained for PUUN (as a marker of infection) and light chains (LC, as markers of B cells) expression by immunofluorescence. The images are representative of experiments performed using two PBMC donors. (**D, E**) PBMCs obtained from patients with PUUV infection were stained for PBs similarly as in Fig 5, except that the Ig-specific Abs were omitted and intracellular staining of PUUN was included using PUUN-specific AlexaFluor647-conjugated Mab 1C12. PUUN$^+$ PBs were detected from CD19$^+$ cells (gated similarly as in Fig 5) by plotting PUUN vs. CD38. (**D**) Representative plots from acute and recovery stage PUUV patient PBMC samples are shown. (**E**) The frequency of PUUN$^+$ cells (above plot) in CD19$^+$ cells (total B cells) and CD19$^+$CD38$^{+++}$ cells (PBs), together with CD38$^{+++}$ cells in CD19$^+$ cells (PBs of total B cells; below plot), were calculated from sequential acute to recovery stage of PUUV-HFRS PBMC patient samples. Means ± standard errors are shown. (**F**) Patient B cells were isolated from PBMCs, spun down on glass slides and fixed, permeabilized and stained for PUUN (as a marker of infection) and light chains (LC, as markers of B cells) expression by immunofluorescence. The scale bars (in **C** and **F**) correspond to 10μm.

immunofluorescence analysis (Fig 6F). Taken together these results suggest that PUUV can infect various B cells subsets *in vitro* and in patients.

## PUUV induces secretion of FLCs from B cells

To assess whether PUUV infection induces B cell activation and FLC production, we inoculated B cells enriched from PBMCs of healthy volunteers with PUUV and UV-PUUV, and mock-infected cells as an additional control. Toll-like receptor 9 (TLR9) ligand CpG served as a positive reference for B cell activation. For comparison, we inoculated a reference set of B cells with Sendai virus (SeV, a potent immune stimulant) and Sindbis virus (SINV, not associated with B cell activation) at multiplicity of infection (MOI) comparable to PUUV inoculum. At day 6 post treatment, we measured $\kappa_{FLC}$, $\lambda_{FLC}$, and whole Ig (A, M and G classes) levels from the B cell supernatants, and quantified the amount of IgA, IgM or IgG secreting B cells (antibody secreting cells; ASCs) by Elispot. PUUV and SeV infections, as well as CpG treatment, induced significantly elevated $\kappa_{FLC}$ and $\lambda_{FLC}$ levels (~2000–4000 vs. ng/l for κ, ~1000–2000 vs. ng/l for γ; Fig 7A) as compared to both mock infection and UV-PUUV inoculation. In addition, the stimulation with CpG and/or the infection with PUUV or SeV all induced, as compared to mock-infected B cells, an approximately two-fold increase in the secretion of soluble IgA (~1–2 vs. 0.8 μg/ml), IgM (1–1.5 vs. 0.75 μg/ml) and IgG (~1.5 vs. 0.75 μg/ml). Similar induction did not occur with UV-PUUV or SINV-inoculated B cells (Fig 7B). Expectedly, based on PB distribution (Fig 5C), PUUV induced ASCs of all the analyzed Ig classes (Fig 7C and 7D). Interestingly, while CpG appeared as more potent inducer of IgM+ and IgG+ ASCs, PUUV showed highest level of IgA+ ASCs (Fig 7D).

To conclude, the data show PUUV infection to activate B cells, which results in a polyclonal Ig response involving not only intact IgA, IgM and IgG, but also $\kappa_{FLC}$ and $\lambda_{FLC}$. The fact that UV-PUUV did not produce a similar effect suggests that B cell activation results from direct viral replication. The observation of increased FLC levels in supernatants of SeV infected and CpG stimulated cells points to FLC secretion to be a typical outcome in B cell differentiation towards ASCs. Curiously, PUUV appeared as the most potent inducer of ASCs secreting IgA.

## Discussion

Hantaviruses can cause either HFRS or HPS when transmitted to humans, and increasing evidence suggests that the pathogenesis of hantavirus infection in humans is due to intense activation of the immune system, rather than direct virus-induced damage. In this study, we show a marked increase in serum $\kappa_{FLC}$ and $\lambda_{FLC}$ levels in both acute HFRS and HPS while the $\kappa_{FLC}$/$\lambda_{FLC}$ ratio was elevated specifically in HFRS. Further studies on HFRS revealed the presence of renal plasma cell infiltration and urinary FLC secretion, which point out towards local, excessive humoral responses. These findings coincided with an increased frequency of circulating PBs in HFRS, a likely source of elevated serum FLCs in both HFRS and HPS. Furthermore, we showed PUUV to infect B cells directly, resulting in proliferation and FLC secretion *in vitro*.

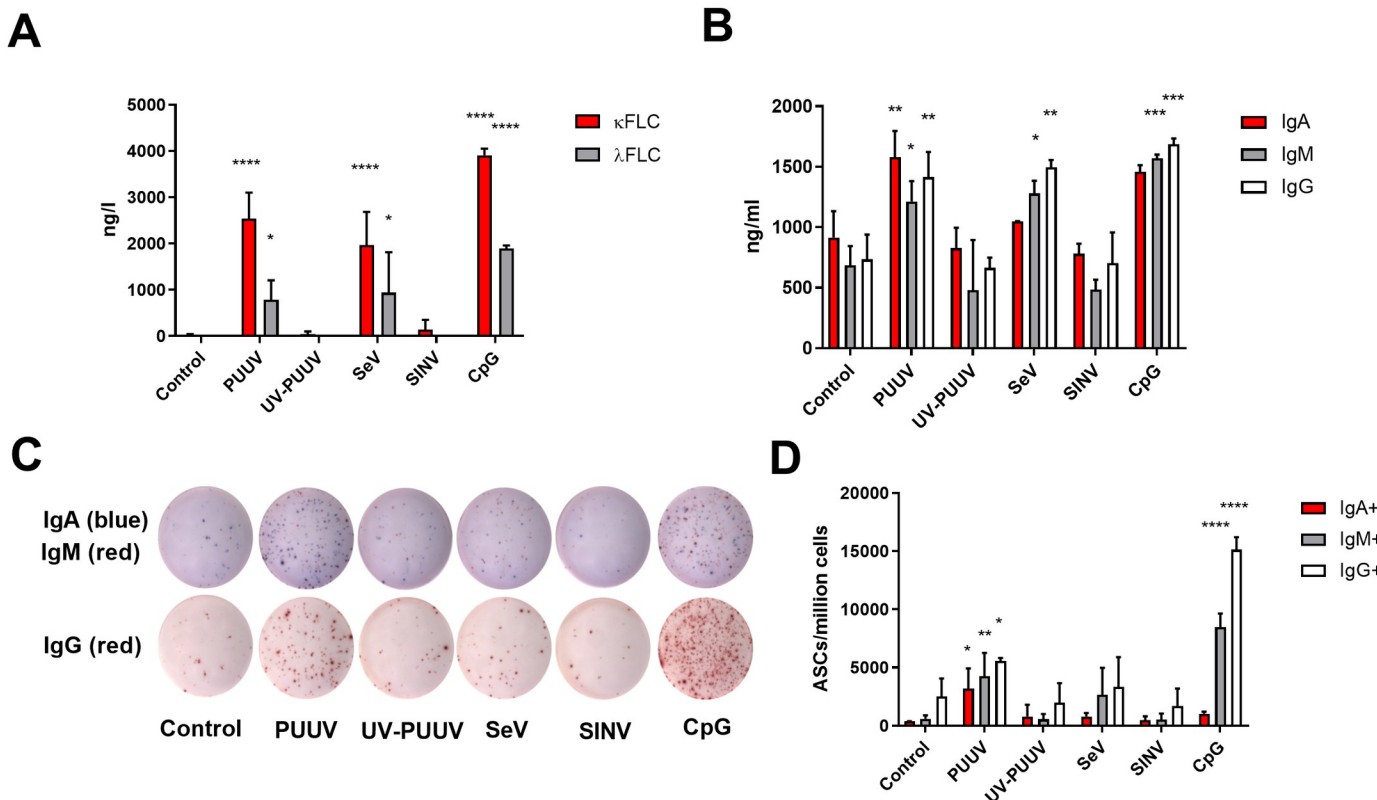

**Fig 7. PUUV activates B cells to produce FLCs and whole Igs.** B cells were enriched from blood of 3 healthy volunteers and infected with live or UV-inactivated purified PUUV, together with Sendai and Sindbis viruses, for comparison (multiplicity of infections for all viruses equals 1). The TLR9 agonist CpG and no treatment were used as a positive and negative controls for B cell activation, respectively. After culturing for 8 days, kappa and lambda FLCs in (A) and IgA, IgM and IgGs in (B) were measured from cell supernatants. Remaining cells were subjected to IgA, IgM and IgG-specific antibody secreting cell (ASC) quantification using Elispot. IgA$^+$ and IgM$^+$ ASCs were simultaneously detected in one well, using AP-conjugated anti-IgA and HRP-conjugated anti-IgM Abs, and IgG$^+$ ASCs in a separate well using HRP-conjugated anti-IgG Ab. Representative Elispot images from one donor are shown in (C), and ASC quantification per input cell number measured from all donors in (D). Significant differences of each treatment group, as compared to non-treated cells, were assessed using two-way ANOVA and Dunnett's multiple comparison tests. $^*$p < 0.05, $^{**}$p < 0.01, $^{***}$p < 0.001.

The findings suggest that B cell activation via hantavirus infection could play a role in the pathogenesis of hantavirus-caused diseases.

The FLC concentrations in hantavirus patient serum exceeded the normal serum reference values: $\kappa_{FLC}$ (3.3–19.4 μg/ml) and $\lambda_{FLC}$ (5.7–26.3 μg/ml) [24]. However, the $\kappa_{FLC}/\lambda_{FLC}$ ratio remained within the normal range (0.26–1.65, [24]) in both acute HFRS and HPS, suggesting that the increase in serum FLCs in acute hantavirus infection is a result of polyclonal B cell activation rather than proliferation of single B cell clone. The kidney mediates FLC clearance, therefore serum FLCs increase as kidney function declines [24]. Thus, the higher $\kappa_{FLC}/\lambda_{FLC}$ ratio in patients with HFRS compared to HPS patients is likely a direct consequence of AKI, further supported by strong positive correlation between serum FLC levels and maximum serum creatinine, an indicator of the severity of AKI.

Clonal FLCs exert renal pathogenesis in multiple ways, including fibril or deposit formations that affect the glomeruli, induce epithelial cell disorders, or cast in the tubular network [25]. Our analysis of PUUV patients' kidney biopsies demonstrated significant infiltration of plasma cells in the renal interstitium. Given that plasma cells produce LCs in excess over heavy chains, secreting the surplus as FLCs [26]: this observation together with the increased urinary FLC levels before the peak in circulating serum FLCs (i.e. lack of correlation between urinary

and serum FLCs), might suggest a local FLC production in the kidneys, which could contribute to AKI in HFRS. While highly speculative, similar plasma cell infiltration and FLC production could occur in the lungs instead of kidneys in acute HPS. Interestingly, increased levels of FLCs associate with different lung disorders [27].

Although previously considered functionally insignificant bystander products of B cell activity, FLCs have recently gained attention as mediators of extrarenal manifestations such mast cell hypersensitivity reactions [28] and influencing the life span of neutrophils [29]. The latter is particularly interesting in terms of hantavirus pathogenesis since neutrophil activation does correlate with disease severity in HFRS [6,30], and local expression of the neutrophil chemoattractant interleukin-8 (IL-8) as well as other neutrophil-related proteins have been observed in patient kidneys [6]. Interestingly, FLCs induce IL-8 and recruit neutrophils to the lungs in murine models of chronic obstructive pulmonary disease [31] and similar phenomena could be envisaged to take place also in kidneys and lungs during acute HFRS and HPS, respectively.

PUUV patient samples collected during the acute phase of illness contained significantly elevated levels of circulating PBs, which returned to normal range in recovery. PBs, present in the blood circulation, are extensively proliferating intermediates of activated B cells, differentiating towards becoming plasma cells [32]. The fact that increased level of circulating PBs coincided with the increased FLC concentration proposes PBs to play a role in their upregulation. We further speculate that the observed plasma cell infiltration to the kidneys might be due to an expansion of the PB population. This PB response in patients led to elevated levels of circulating total IgA, IgM and IgG during acute and early recovery stages. The PBs collected in the acute and recovery phases did not show significant differences in the frequencies of IgA$^+$, IgG$^+$ or κLC$^+$ cells. However, the IgM$^+$ PBs frequency diminished towards the late acute/early recovery period, suggesting class-switched PBs to dominate at the later stages of the disease. We conjecture that, in addition to insufficient renal clearance, PBs/plasma cells account for the elevated FLC levels in the circulation of patients with acute PUUV-HFRS. A massive and transient circulating PB response also occurs in acute ANDV-caused HPS [12]. The ANDV infection-induced PBs showed specificity towards both viral and host antigens, suggesting a polyclonal bystander B cell activation. Along the same idea, we have earlier shown that EBV-specific memory T cells are boosted after PUUV hantavirus infection [33] and recently showed heterologous boosting of nonrelated immunity during acute PUUV [34], indicating that immune responses towards hantaviruses are not entirely specific. Garcia et al. reported also the HPS patients to show a significantly elevated frequency of IgA$^+$ PBs [12], whereas our results display similar levels of IgA$^+$ PBs in PUUV patients.

Analysis of PUUV patients' PBMCs revealed PBs as positive for PUUV antigen, with the fraction of positive PBs waning down towards convalescence. The finding suggests productive infection of B cells that could drive their differentiation into PBs. Supporting the hypothesis, we found purified B cells to be permissive for PUUV infection *in vitro*, leading to B cell proliferation and secretion of both intact Igs and FLCs. Thus, the findings suggest that the observed PB/plasma cell response in acute PUUV-HFRS to be a T cell-independent (TI) "innate-like" response, rather than T cell-dependent (TD) class-switched virus-specific response that occur in germinal centers. If this is the case, the virus-induced activation of B cells is highly polyclonal rather than solely virus-targeted, and indeed Garcia et al. found PB responses towards non-virus-related antigens in acute HPS [12]. An overt non-specific polyclonal B cell response could explain the massive FLC production seen in HFRS and HPS.

Acute polyclonal PB response occurs also in other viral infections (reviewed by [35]) including those caused by dengue flavivirus (DENV) [36]. The pathogenesis of DENV bears similarities to hantavirus diseases, in which vascular leakage and hemorrhages are key clinical

findings, while renal insufficiencies are also common, although less reported [37,38]. The PB response in acute DENV can be as high as 80% of all CD19[+] B cells, constituting 30% of total PBMCs [39] and associating with disease severity [40]. Similarly as shown in the current study for PUUV, B cells can support DENV infection *in vitro* that results in B cell activation and production of polyreactive Abs [41], with the majority of DENV RNA within patient PBMCs being associated with naïve B cells [42]. Taken together, it seems that the role of FLCs in the pathogenesis of hemorrhagic fevers, and infectious diseases in general, is not yet fully recognized.

## Supporting information

**S1 Fig. Sequential patient samples used in this study.** The number of samples at each day post onset of fever are indicated for PBMC, serum and urine obtained from PUUV-caused HFRS patients.
(TIF)

**S2 Fig. Correlograms of clinical and laboratory parameters in acute hospitalized PUUV-caused HFRS.** Spearman correlation analysis of serum κFLC, λFLCs and their ratio (in **A**) and κ- and λ-LC specific plasma cells (PC) in kidneys (in **B**) with clinical and laboratory parameters are shown. The color of the circles indicates the value of Spearman correlation coefficients as depicted in the legend on the right and increasing size of the circles indicate decreasing p-value (statistical significance is reported as * = $p < 0.05$, ** = $p < 0.01$, *** = $p < 0.001$ and **** = $p < 0.0001$). Creatinine, leukocytes, C-reactive protein (CRP), interleukin (IL)-6 and thrombocytes correspond to maximum and minimum values in blood during hospital stay, respectively. Hosp. LOS = Length of stay in hospital. cUAlb = Overnight urinary albumin excretion. CD68, CD14, CD16 and HLA-DR correspond to the extent infiltration of CD68[+], CD14[+], CD16[+] and HLA-DR[+] leukocytes in kidneys.
(TIF)

**S3 Fig. Association of serum FLCs with disease severity in hantavirus-caused diseases.** (**A**) Serum κ and λ FLC levels in PUUV-caused HFRS patients with and without the need of dialysis (n = 26 and n = 4) and (**B**) ANDV-caused HPS patients stratified based on disease severity (1 = with prodromal symptoms without respiratory involvement; 2 = mild to moderate respiratory compromise without hemodynamic compromise; 3 = with severe respiratory insufficiency with hemodynamic compromise; 4 = with severe respiratory insufficiency with refractory-to-treatment hemodynamic compromise, with a final fatal outcome). Statistically significant differences assessed with Mann-Whitney test and reported as * = $p < 0.05$. The bars indicate medians + interquartile ranges.
(TIF)

**S4 Fig.** The concentration of FLCs (κ in **A** and λ in **B**) in urine samples collected at indicated days post onset of fever from patients with acute PUUV-HFRS (n = 13) were correlated to total urinary protein levels using Spearman's rank correlation coefficient. Non-linear association between parameters is depicted by the red line.
(TIF)

**S5 Fig. Gating strategy for the detection of plasmablasts in the circulation of patients with acute PUUV-HFRS by multi-parametric flow cytometry.** After gating on PBMCs (SSC-A vs. FSC-A), single cells (FSC-H vs. FSC-A) and CD19+ B cells (CD3, CD14, CD56, CD66 vs. CD19), were identified as CD27+CD38++ cells (CD27 vs. CD38). IgM+ and IgA+ PBs were gated from the total PB fraction (IgM vs. IgA) and IgG+ PBs (SSC-A vs. IgG) from the

IgM-IgA- fraction. Representative plots for an acute and recovery stage PUUV-HFRS are shown.
(TIF)

**S6 Fig. Replication of PUUV in B cells.** B cells were isolated from healthy volunteer PBMCs by negative selection and a representative histogram of the percentage of CD19+ B cells in the isolated fraction as assessed by flow cytometry is shown in (**A**). (**B**) Isolated B cells were left untreated, activated with CpG (1 μM) or infected with live (5 FFFU / cell) or UV-inactivated PUUV and subjected to non-reducing western blot at 5 days post infection. The viral N protein and κLC expressions (both green) were detected by specific Abs produced in mouse or goat, respectively. A rabbit Ab to actin (red) served as a protein loading control. Free and heavy chain (HC)-associated light chains (LC) are indicated.
(TIF)

**S1 Table. Individual PUUV-HFRS patient characteristics from which serum samples at 1ˢᵗ day of hospitalization was included in this study.**
(DOCX)

**S2 Table. Individual ANDV-HPS patient characteristics from which serum samples were included in this study.**
(DOCX)

**S3 Table. Individual PUUV-HFRS and control patient characteristics from which kidney biopsies were included in this study.**
(DOCX)

## Acknowledgments

We thank Sanna Mäki, Johanna Martikainen and Mira Utriainen for expert technical assistance.

## Author Contributions

**Conceptualization:** Jussi Hepojoki, Satu Hepojoki, Lauri Kareinen, Tomas Strandin.

**Data curation:** Jussi Hepojoki, Luz E. Cabrera, Carla Bellomo, Valeria Martinez, Tomas Strandin.

**Formal analysis:** Jussi Hepojoki, Luz E. Cabrera, Carla Bellomo, Valeria Martinez, Tomas Strandin.

**Funding acquisition:** Jussi Hepojoki, Satu Hepojoki, Antti Vaheri, Jukka Mustonen, Tomas Strandin.

**Investigation:** Jussi Hepojoki, Carla Bellomo, Tomas Strandin.

**Methodology:** Jussi Hepojoki, Leif C. Andersson, Tomas Strandin.

**Project administration:** Jussi Hepojoki, Tomas Strandin.

**Resources:** Antti Vaheri, Satu Mäkelä, Jukka Mustonen, Olli Vapalahti, Valeria Martinez, Tomas Strandin.

**Supervision:** Jussi Hepojoki, Tomas Strandin.

**Validation:** Leif C. Andersson, Tomas Strandin.

**Visualization:** Jussi Hepojoki, Luz E. Cabrera, Tomas Strandin.

**Writing – original draft:** Jussi Hepojoki, Tomas Strandin.

**Writing – review & editing:** Jussi Hepojoki, Antti Vaheri, Satu Mäkelä, Jukka Mustonen, Olli Vapalahti, Valeria Martinez, Tomas Strandin.

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
