## [Decision Letter · Decision Letter 0]

24 May 2021

Dear Mr. Strandin,

Thank you very much for submitting your manuscript "Hantavirus Infection-Induced B Cell Activation Elevates Free Light Chains Levels in Circulation" for consideration at PLOS Pathogens. As with all papers reviewed by the journal, your manuscript was reviewed by members of the editorial board and by several independent reviewers. In light of the reviews (below this email), we would like to invite the resubmission of a significantly-revised version that takes into account the reviewers' comments.

We cannot make any decision about publication until we have seen the revised manuscript and your response to the reviewers' comments. Your revised manuscript is also likely to be sent to reviewers for further evaluation.

Sincerely,

Tony Schountz, PhD

Guest Editor

PLOS Pathogens

Sonja Best

Section Editor

PLOS Pathogens

Kasturi Haldar

Editor-in-Chief

PLOS Pathogens

orcid.org/0000-0001-5065-158X

Michael Malim

Editor-in-Chief

PLOS Pathogens

orcid.org/0000-0002-7699-2064

Reviewer's Responses to Questions

**Part I - Summary**

Reviewer #1: This manuscript builds on a recent publication by the same authors, where they reported PUUV-N protein binding FLCs in PUUV-infected patients. In this follow-up study Hepojoki and colleagues show a general increase of FLCs in blood of HFRS and HPS patients. During acute HFRS they observe increased FLC levels in blood and urine, and correlations between FLC levels with markers of kidney failure. Further, they show that PUUV infect B cells, which lead to secretion of FLCs. The findings are novel in a hantavirus perspective and add to previous reports of FLC secretion from activated B cells.

A strength of the study is the inclusion of HFRS and HPS patient samples, and the exploration of a potentially novel finding that can play a role in hantavirus pathogenesis.

A weakness of the study is that although the authors report a correlations between levels of FLCs and kidney damage, the potential role of FLCs in HFRS/HPS pathogenesis is unclear – does secreted FLCs represent more of a general response to immune activation (not specific for HFRS)? Moreover, an important finding is the potential infection of B cells. While the presented data do strongly suggest that B cells are infected, further in vitro analysis, with more specific methods, are needed to conclude this.

Reviewer #2: This study demonstrates some interesting observations of the increased presence of FLCs during PUUV infections. The science in sound however the major limitation is that this is merely an observational study. The authors should include a comparison group with severe HFRS disease, for example patients diagnosed with HTNV or DOBV infections. Instead, they refer to PUUV-infections as HFRS, which they technically are; however due to the mild presentation have been most commonly referred to PUUV cases as NE and HFRS is reserved for the more severe Old World hantavirus infections. If the authors could demonstrate differences in non-hantavirus infections vs PUUV vs HTNV infections this manuscript would be far more informative.

Reviewer #3: Overall, this is a well written manuscript, with a strong experimental method, clear results and figures, and well thought out discussion. The data presented here is building on previously published observation regarding free light chains (FLCs) as diagnostic analytes for Hantavirus infections, more specifically examining the infection-mediated production of FLCs and how their increase is related to pathology.

There are numerous strengths to this manuscript. The use of human clinical samples taken from acute PUUV infections is particularly valuable in this context, as is the sequential data taken during the course of the infection and subsequent recovery period. The extremely in-depth and comprehensive analysis of the roles of FLCs is also impressive, with multiple experimental techniques employed. The experimental question regarding the role of B cell activation and FRCs levels related to acute PUUV disease is very well answered.

One weakness is the broad capture of all acute PUUV cases (and ANDV cases as well for the HPS data) under one umbrella. I would have liked to have seen a more granular breakdown in some of the data analysis to examine whether variables (if the researchers have access) such as sex, age of patient, viral load, antibody responses etc. affected the parameters measured here. Even if the data presented here is independent of these factors, that should be stated somewhere.

**Part II – Major Issues: Key Experiments Required for Acceptance**

Reviewer #1: 1. The finding of increased FLCs in circulation, and correlations to AKI, is interesting. However, to better understand potential mechanisms behind severity of HFRS/HPS, especially if FLCs play a role in pathogenesis, it would be interesting to compare levels of FLC in HPS-patients with fatal outcome vs survivors, and for HFRS-patients analyse for possible correlations to severity (for instance IL-6 levels and/or other markers of severity). Including these analysis would strengthen a possible role for FLC in pathogenesis/severity during hantavirus infection.

2. Figure 2. In figures 2C and 2D mean is shown, however, there is a large fluctuation in the values between samples and between time points, hence outliers can have a strong effect. I suggest showing median or geometric mean instead of mean. Further, in figures 2C and 2D, or in the legend, include number of patients analysed at the different time-points.

Does the k/l FLC ratio correlates to creatinine levels in HFRS-patients?

Are there data available for creatinine in HPS-patients? A comparison also on HPS patient data could indicate if the same pattern is observed in HFRS and HPS, or if it is unique for HFRS.

3. Infection of B cells with PUUV. In figure 6A PUUV N pos cells are shown. However, the staining for PUUV N+ cells is not very distinct. Histogram showing PUUV N could be an alternative way to show this data.

In figure 2C one PUUV N positive CD19+ cell is shown. Please include a figure with data on frequency of infected cells. In lines 123-125 it is stated that the enriched B cells were sorted to 100% purity. How was this confirmed? If possible, please provide as a supplementary figure flow data showing purity after B cell enrichment. My concern is that if very few infected cells were observed, they might represent other cell types than B cells.

While the presented data strongly suggest B cells are infected, clear evidence is lacking. Additional assays are needed to conclude that B cells are successfully infected. This can be achieved by analysing for instance progeny virus production and/or levels of viral RNA over time which could confirm that B cells can be infected by PUUV, and/or support replication.

If possible, it would also be interesting to see if ANDV can infect B cells, and if ANDV infection has similar effects as PUUV or not on FLC secretion.

Reviewer #2: In this reviewers opinion a group of HTNV or DOBV -HFRS is required for more appropriate comparisons.

Reviewer #3: As mentioned previously, some further breakdown within the HFRS patients based on other variables would increase the strength of some aspects of this manuscript (if data is readily available). This would especially relate to the data generated for figure 1, 2, 3 and possibly 4.

**Part III – Minor Issues: Editorial and Data Presentation Modifications**

Reviewer #1: Lines 65-66. It is rather well established that the route from lungs to kidneys is explained by systemic spread of hantaviruses: the main target of hantaviruses are endothelial cells lining capillaries, and hence the virus can spread to all organs in the body.

Lines 104-105. Which k and l LC-specific antibodies were used?

Lines 118-119. The link to Milteny can be omitted.

Lines 124-125.

Line 149. From which company is the secondary Ab?

Line 162. What conditioned medium was used?

Figure 1. It is difficult to see the bar for interquartile ranges for HPS-patients.

Lines 222-223. This sentence seems a little bit misplaced, maybe move to the next section?

Line 241. This sentence is rather speculative, please rephrase.

Figure 3. Lines 244-247. Please provide a supplementary figure showing the correlations, they are not obvious when looking at figure 3A.

Lines 325-326. This sentence is very speculative, the authors have not shown that PUUV infection lead to activation of B cells to become PBs.

Figure 6C. The size bar should be explained in the figure legend

Reviewer #2: (No Response)

Reviewer #3: -For clarity, somewhere in the first results paragraph it should be stated that the HPS serum comes from ANDV-infected patients. This should also be in the figure legend.

-In Figure 2, it would be useful (if possible) to indicate what or where humans “normal” values for the measured analytes are.

-How were serum creatinine levels determined? This should be in the materials and methods.

PLOS authors have the option to publish the peer review history of their article (what does this mean?). If published, this will include your full peer review and any attached files.

Reviewer #1: No

Reviewer #2: No

Reviewer #3: No
---

## [Decision Letter · Decision Letter 1]

27 Jul 2021

Dear Mr. Strandin,

We are pleased to inform you that your manuscript 'Hantavirus Infection-Induced B Cell Activation Elevates Free Light Chains Levels in Circulation' has been provisionally accepted for publication in PLOS Pathogens.

Best regards,

Tony Schountz, PhD

Guest Editor

PLOS Pathogens

Sonja Best

Section Editor

PLOS Pathogens

Kasturi Haldar

Editor-in-Chief

PLOS Pathogens

orcid.org/0000-0001-5065-158X

Michael Malim

Editor-in-Chief

PLOS Pathogens

orcid.org/0000-0002-7699-2064

Reviewer Comments (if any, and for reference):

Reviewer's Responses to Questions

**Part I - Summary**

Reviewer #1: My previous questions and comments have been adequately answered. Overall the work is well done and conclusions appropriate.

Reviewer #3: Authors have done a nice job addressing all comments and concerns in the revised manuscript. No suggestions for major or minor comments. All previous suggestions were incorporated well into the report.

**Part II – Major Issues: Key Experiments Required for Acceptance**

Reviewer #1: (No Response)

Reviewer #3: (No Response)

**Part III – Minor Issues: Editorial and Data Presentation Modifications**

Reviewer #1: (No Response)

Reviewer #3: (No Response)

PLOS authors have the option to publish the peer review history of their article (what does this mean?). If published, this will include your full peer review and any attached files.

Reviewer #1: No

Reviewer #3: No

---

## [Editor Report · Acceptance letter]

5 Aug 2021

Dear Mr. Strandin,

We are delighted to inform you that your manuscript, "Hantavirus Infection-Induced B Cell Activation Elevates Free Light Chains Levels in Circulation," has been formally accepted for publication in PLOS Pathogens.

Best regards,

Kasturi Haldar

Editor-in-Chief

PLOS Pathogens

orcid.org/0000-0001-5065-158X

Michael Malim

Editor-in-Chief

PLOS Pathogens

orcid.org/0000-0002-7699-2064